

# Link between the Surface Mass Balance of the Greenland Ice Sheet and the North Atlantic Oscillation under preindustrial and last interglacial climates: a study with a Coupled Global Circulation Model

Silvana Ramos Buarque[1] and David Salas y Melia[1]

[1]Centre National de Recherches Météorologiques (CNRM), Toulouse, 31057, France

*Correspondence to*: S. Ramos Buarque (silvana.buarque@meteo.fr)

**Abstract.** The relationship between the Surface Mass Balance (SMB) of the Greenland Ice Sheet (GrIS) and the North Atlantic Oscillation (NAO) is examined from numerical simulations performed with a new atmospheric stretched grid
configuration of the Centre National de Recherches Météorologiques - Coupled Model (CNRM-CM) version 5.2 under three periods : preindustrial climate, a warm phase (early Eemian, 130ka BP) and a cool phase (glacial inception, 115 ka BP) of the last interglacial. The horizontal grid of the atmospheric component of CNRM-CM5.2 is stretched from the tilted pole on the Baffin Bay (72°N, 65°W) in order to obtain a higher spatial resolution on Greenland. The correlation between simulated SMB anomalies averaged over Greenland and the NAO index is weak in winter and significant in summer (about 0.6 for the
three periods). In summer, spatial correlations between the NAO index and SMB display different patterns from one period to another. These differences are analysed in terms of the respective influence of the positive and negative phases of the NAO on accumulation and melting. Accumulation in South Greenland is significantly correlated with the positive (negative) phase of the NAO in a warm (cold) climate. Under preindustrial and 115 ka climates, melting along the margins is more correlated with the positive phase of the NAO than with its negative phase, whereas at 130 ka it is more correlated with the
negative phase of the NAO in North and North-East Greenland.

## 1 Introduction

The recently observed acceleration of mass loss from the Greenland ice sheet (Hanna et al., 2013; Fettweis et al., 2013; Gillet-Chaulet et al., 2012 and references there in) is a concern due to its possible contribution to future sea-level rise. For example, Yan et al. (2014) estimated the GrIS contribution to global sea-level rise by 2100 by means of ice-sheet model
simulations (including dynamics) forced with output from 20 CMIP5 (Coupled Model Intercomparison Project phase 5) to range from 0 to 16 (0 to 27) cm under the Representative Concentration Pathways (RCP) 4.5 (RCP8.5). For a given RCP, this uncertainty is mainly due to the large spread among CMIP5 model simulations. Furthermore, Fürst et al. (2015) found that the largest source of uncertainty in projections of the GrIS contribution to sea-level rise arises from the SMB rather than from the dynamics of the ice sheet.



The SMB of the GrIS has shown a downward trend since the early 1990s (Ettema et al., 2009). This downward trend is due to increased surface melting (Sasgen et al., 2012; Vernon et al., 2013). For example, during 12-15 July 2012, surface melting affected over 97% of the GrIS (Nghiem et al. 2012; Dahl-Jensen et al., 2013), in the context of a negative phase of the North Atlantic Oscillation (NAO). Typically, this weather regime is associated with an anticyclonic circulation centered over Greenland that induces warmer and drier summers than normal and southerly warm air advection along the western

Greenland coast at the surface and at 500 hPa. In recent years, changes in atmospheric circulation explain about 70% of the summertime warming in Greenland (Fettweis et al., 2013). Over the last 30 years, changes in the NAO index were found in winter and summer but not in spring and autumn (Hanna et al., 2013). In winter, the year-to-year climate variability of the North Atlantic region is well captured by the NAO index because the atmospheric circulation is active and well organized. By contrast, in summer, the NAO explains a smaller fraction of the circulation variability in this region (Folland et al., 2008).

On top of NAO changes, Arctic amplification plays a role in the recent SMB trend. Pithan and Mauritsen (2014) quantified the contributions of various feedbacks to Arctic temperature amplification from CMIP5 models and found that the largest contribution is due to a surface temperature feedback, because of the smaller increase in surface outgoing longwave radiation per °C of warming at cold surface temperatures than at higher temperatures prevailing at lower latitudes.

In this paper, we focus on the link between NAO and SMB and its components (accumulation and melting) and its stability

under different climates. In particular, the early and late periods of the last interglacial state (Eemian) respectively correspond to a warm and a cold climate, which can, to some extent, serve as analogs to interpret recent and future climate changes. Mechanisms of such changes can be studied by using Coupled Global Circulation Models (CGCMs). However, current CGCMs, that couple atmosphere-land surface and ocean-sea ice models are increasingly comprehensive, but their typical horizontal resolution is currently around 100 km, which is too coarse to correctly represent local circulation in

Greenland and surface moisture flux convergence. For example, snow sublimation is generally underestimated in CGCMs because the realism of this process highly depends on a good representation of the wind and especially on its maxima which increase with resolution (Lenaerts et al., 2012). Ettema et al. (2009) have quantified SMB on the GrIS by using high-resolution (about 11 km) limited-area regional climate model simulations and found that considerably more mass accumulates than previously thought, revising upwards earlier estimates by as much as 63%. This result points out the need

to use high resolution models for estimating SMB. High resolution is also a necessary condition to well capture the spatial variability of the snow melt on margins of the GrIS especially where snow melt gradients are strong. This ability becomes all the more important as the expected trend of SMB in a warming climate is an enhanced melting along the GrIS margins. Hence, in order to locally increase horizontal resolution at a reasonable computational cost, in this study we use a stretched grid configuration (with enhanced resolution over Greenland) of the atmospheric component ARPEGE-Climat of CNRM-

CM.

The questions addressed in this paper are: **i)** What is the link between NAO and the variability of the GrIS SMB under preindustrial climate ? **ii)** How robust is this link under the warm and cool phases of the Eemian ? **iii)** What are the regions where SMB is most influenced by the NAO and to what extent ? This paper is structured as follows. Section 2 describes the



stretched grid configuration of CNRM-CM and the experimental design for this study. The preindustrial control simulation

performed with CNRM-CM is analysed in section 3 and compared with the ECMWF Reanalysis ERA-Interim (Dee et al., 2011) and a previous CMIP5 simulation. The SMB and its link with NAO as simulated by CNRM-CM are compared with a simulation performed with MAR (Modèle Atmosphérique Régional). Section 4 is devoted to assessing the response of Greenland climate to large scale changes under the warm (130 ka) and the cool (115 ka) phases of the Eemian, with a focus on summer.

**2 Features of the climate modelling simulations**

**2.1 Modelling tool**

This study uses the CGCM CNRM-CM5.2 developed jointly by CNRM (Centre National de Recherches Météorologiques) and CERFACS (Centre Européen de Recherche et de Formation Avancée en Calcul Scientifique) as described by Voldoire et al. (2013). The components of CNRM-CM5.2 are the atmospheric model ARPEGE-Climat (Deque et al., 1994), the surface

platform SURFEX (Le Moigne et al., 2009), the river routing TRIP (Oki and Sud, 1998), the ocean model NEMO (Madec, 2008) and the sea ice model GELATO (Salas y Mélia, 2002). The components of CNRM-CM5.2 are coupled by means of the OASIS coupler (Valcke, 2006).

The ice mass transport due to the dynamics of the GrIS is not explicitly represented within CNRM-CM5.2. To circumvent this, the GrIS is represented by an initially prescribed huge amount of snow that evolves according to the balance between

the snowfall rate, the direct sublimation and the snow melt, but without any modification of the topography of Greenland, the snowpack being represented by the one-layer snow scheme of Douville et al. (1995). To avoid unrealistic snow accumulation on the GrIS and an associated decrease in the modelled sea level, a pseudo-calving flux is computed at every time step from the spatially integrated snow reservoir excess over the GrIS and is distributed over the ocean north of 60°N.

The atmospheric component ARPEGE-Climat is used in a ''low-top'' configuration with 31 vertical levels (the highest level

is set at 10 hPa). The horizontal grid is defined by a T127 spectral triangular truncation (a global mean spatial resolution of about 150 km). In this study, however, we chose to increase horizontal resolution to 40-50 km over Greenland in order to improve the spatial representation of SMB, in particular near the GrIS margins. To do so, the north pole of the ARPEGE-Climat horizontal grid was displaced to the Baffin Bay (72°N, 65°W) and the grid was stretched by a factor of 2.5 following the spherical harmonic-based functions on a transformed sphere (Courtier and Geleyn, 1988). In the rest of this study, this

configuration of CNRM-CM5.2 will be referred to as NPS (North Pole Stretched). Different previous studies have used this functionality of increasing the horizontal resolution in a region of interest while decreasing it in other regions without any additional computational cost compared to a globally uniform resolution (e.g. Lorant and Royer, 2001 ; Doblas-Reyes, 2002 ; Chauvin et al., 2006). The physics and the calculations of the non-linear terms require spectral transforms onto a reduced Gaussian grid (Hortal and Simmons, 1991).

The ocean component is deployed on the horizontal quasi-isotropic tripolar grid ORCA1 (Hewitt et al. 2011) with 42 vertical levels and a horizontal resolution of about 1°. This grid has a latitudinal grid refinement of 1/3° at the Equator, andn the North Pole singularity is replaced by two poles located in Canada and Siberia.

## 2.2 Experimental Set-up

Three 280-year simulations were performed with NPS : preindustrial (NPS-0k), early Eemian climate (130 ka BP, denoted as
NPS-130k) and glacial inception (115 ka BP, denoted as NPS-115k). These simulations differ only by the astronomical parameters (orbital eccentricity, axial tilt or precession and obliquity) that drive incoming insolation changes (Berger, 1988). In this study, we defined these parameters following Berger (1978) [see Table 1] . In all the simulations, the concentrations of tropospheric aerosols (organic and black carbon, sea salt, sulphate and sand dust) are estimates from the LMDz-INCA chemistry-climate model (Szopa et al., 2013) for years 1850-1860, considered as representative of preindustrial conditions.
The atmospheric concentrations of well-mixed greenhouse gases (carbon dioxide, methane, nitrous oxide, ozone and CFCs) are yearly means for 1850. The 3D stratospheric ozone concentration is averaged from years 160-259 of the CMIP5 preindustrial control experiment run with CNRM-CM5.1. The solar constant is equal to 1365.6537 W/m2 for all the experiments, and the concentration of stratospheric aerosols produced by volcanic eruptions is a monthly zonal mean climatology derived from Ammann et al. (2003).
Atmospheric state variables (temperature, pressure, humidity and wind fields) were initialized from a previous forced integration of ARPEGE-Climat simulation. The initial states of NEMO and GELATO correspond to the first year of the CMIP5 preindustrial control experiment run with CNRM-CM5.1.

## 3 Evaluation of the preindustrial control integration

The NPS-0k simulation was integrated for 280 years without discarding the spin-up since the model reaches a steady state
soon after initialization. This is probably due to the fact that NPS-0k and CMIP5 preindustrial simulations essentially differ by their atmospheric horizontal grids. In the rest of this study, all the analyses of NPS will be based on the entire simulation.

### 3.1 Model evaluation

Differences between the simulated time-mean 2m air temperature in the preindustrial experiments NPS-0k and CMIP5 (years 1-280) and the ERA-Interim reanalysis for winter (DJF) and summer (JJA) over the 1979-2012 period are plotted in Fig. 1.
Note that these differences do not only depict model biases but also include the climate change signal since the preindustrial era.
The near surface warm biases along the eastern boundary currents of Peru and Benguela seen in the preindustrial CMIP5 simulation are more pronounced for NPS-0k. These well-known biases are due to the poor representation of coastal ocean





upwelling and strato-cumulus, leading to an overestimated surface downwelling short-wave (Voldoire et al., 2013). A weak

(CMIP5) to moderate (NPS-0k) warm bias can be seen in DJF in the Southern Ocean. This bias is probably due to several coupled processes. A rough representation of turbulent surface heat and momentum fluxes and vertical turbulent mixing in the ocean and atmospheric boundary layers, particularly in their entrainment zones, could be the causes, among others, of such biases. Biased kinetic energy transfers at the air-sea interface is also a potential source of oceanic biases because this coupled ocean-atmosphere process is particularly active in regions of strong currents (Giordani et al., 2013). Previous

simulations performed with a low horizontal resolution configuration of CNRM-CM suggest that these biases are probably amplified by the much coarser horizontal resolution of the stretched model in this region (around 300 km) than in Baffin Bay.

The North Atlantic cold bias (off Newfoundland) is common to all coupled climate models using NEMO in its ORCA-1° configuration. For NPS-0k, the bias is similar in DJF and JJA whereas for CMIP5 it is more pronounced in DJF than in JJA.

The Arctic is also dominated by a cold bias that is more pronounced in CMIP5 than NPS-0k in winter. The cold winter bias in the Barents Sea already existed in CNRM-CM3 and remains in CNRM-CM5.2. Even if the geographical distribution of Arctic sea ice is generally well simulated by CNRM-CM5.2, particularly in winter, the ice edge in the Greenland and Barents seas does not match observations well.

In order to evaluate accumulation on the GrIS, the annual mean and monthly mean (January and July) precipitation

simulated for NPS-0k, CMIP5 and ERA-Interim are plotted in Fig. 2. The simulated solid precipitation strongly depends on the model resolution, especially along the southeastern Greenland coast where the topography varies sharply over short distances and acts as a barrier for the atmospheric flow. The seasonal variations of precipitation over South and North Greenland are out of phase, with annual maximum values occurring respectively in January and July. In January, the mean precipitation in NPS-0k along the southeastern and southwestern Greenland coasts is similar to ERA-Interim. The

improvement due to the higher horizontal resolution of NPS-0k compared with CMIP5 is clear for the representation of the highest annual precipitation (higher than 0.8 mWE /yr) and of the distribution of precipitation along the coast. In July, the simulated precipitation in NPS-0k over South Greenland and along the western margin of the GrIS is very similar to ERA-Interim. Note that the most important contribution to the annual total precipitation is from July precipitation.

The NAO index can be defined as the difference in sea-level atmospheric pressure between Lisbon (Portugal) or Ponta

Delgada (Azores) and Stykkisholmur or Reykjavik (Iceland) (Hurrel, 1995). The drawback of this proxy is that it does not account for the fluctuations of the locations of the Icelandic low and the Azores high. This implies that the NAO station-based index does not completely capture the seasonal, interannual and multidecadal spatial variability of the North Atlantic pressure patterns (Hanna et al., 2013). The NAO index can also be defined is as the leading Principal Component (PC) of atmospheric pressures usually at sea level, 850hPa or 500hPa. The associated empirically-determined orthogonal function

(EOF) provides the spatial structure of NAO (Bjornsson and Venegas, 1997).

In this work, the NAO index is defined as the normalized PC associated with the first EOF (EOF1) of the detrended monthly sea-level pressure (SLP) anomalies in the North Atlantic (20°N–70°N; 90°W–40°E). Fig. 3 shows the EOF1 for NPS-0k and



ERA-Interim (1979-2016) in winter (DJF) and summer (JJA). In DJF, the positions of the simulated centers of action of NPS-0k are similar to those of ERA-Interim. In JJA, only the southern center of action in NPS-0k reveals a slight
southwestward shift compared to ERA-Interim. The EOF1 of NPS-0k and ERA-Interim explain respectively 35.9% and 50.3% of the total variance of SLP in DJF and 30.3% and 36.2% in JJA.

**3.2 Simulated SMB mean-states**

The SMB can be written as:

$$SMB = P - E - M \quad , \tag{1}$$

where P, E and M (all positive) respectively represent snowfall, the sublimation of the snowpack and surface melting. Fig. 4 compares accumulation C=P-E, melting M and SMB diagnosed from NPS-0k with their counterparts simulated by MAR for the period 1979-2012, which serves as a reference. The latter simulation was performed with MAR version 3.2 (Fettweis et al., 2013) at a horizontal resolution of 25 km and was driven by ERA-Interim at its lateral boundary conditions.

NPS-0k and MAR accumulations compare well and show that surface elevation strongly influences accumulation. Three
regions of accumulation were identified in these simulations. A "dry" region in central and North-East Greenland, where C is less than 0.2 m/yr, a "wet" region along the southeastern and southwestern margins of the GrIS where C is greater than 1 m/yr and the rest of Greenland where accumulation is intermediate. NPS-0k reproduces quite well the "wet" zone simulated by MAR thanks to its relatively high resolution. The "dry" zone is also well simulated, even if in the central part of South Greenland the net accumulation is less in NPS-0k than in MAR. In central Greenland, the lower annual mean accumulation
in NPS-0k compared to MAR is due to the greater summertime sublimation in NPS-0k than in MAR.

GrIS-averaged seasonal accumulation, melting and SMB for NPS-0k and MAR are presented in Table 2. These spatial integrations were computed after interpolating output from both models on an rectilinear grid and masked over the same exogenous Greenland mask obtained from the Global Database of Administrative areas (GADM, http://www.gadm.org/country). In DJF, the mean accumulation averaged on the GrIS in NPS-0k is in close agreement with
MAR (0.31 and 0.32 mWE/yr respectively), whereas in JJA, it is lower in NPS-0k than in MAR (0.29 and 0.34 mWE/yr respectively to NPS-0k and MAR).

Regarding melting, there is less agreement between NPS-0k and MAR. In NPS-0k, the simulated melting rates are underestimated along the margins, especially in the southwestern part of the GrIS, south of the Jakobshavn region, and along the northeastern part of the GrIS. MAR displays much higher melting rates along the margins. Melting is underestimated in
NPS-0k mainly because in CNRM-CM5.2 the minimum albedo of permanent ice is set to 0.8, which hampers the feedback between albedo, solar radiation absorption and melting. Moreover, even at a horizontal resolution of 50 km, the relatively steep topography of the GrIS near the margins cannot be correctly represented, which probably contributes to the biased simulated surface melting in this area.





Melting exceeds accumulation near the margins of the GrIS. This feature is typical of a warm climate but is not simulated by
NPS-0k. In the "dry" region, NPS-0k and MAR display slightly positive SMBs (<0.2 mWE/yr). All in all, the SMB is
reasonably represented in the NPS-0k simulation compared to the reference MAR, even if the surface melting of the
snowpack is strongly underestimated near the margins of the GrIS.

### 3.3 Relationship between the interannual variability of GrIS SMB and NAO

The NAO index explains much of the weather and climate variability over the North Atlantic and Greenland, hence in this
section we examine its link with the variability of GrIS SMB. We compute the NAO index from ERA-Interim since this
reanalysis is the lateral boundary condition of the MAR regional simulation that we used as a reference to validate SMB and
its components. Fig. 5 shows the GrIS-averaged SMB simulated by MAR (1979-2012) and the NAO index from ERA-
Interim. These reconstructions for the 1979-2012 period will serve as a basis for comparison with our model simulations.
The correlation between the two time-detrended variables is -0.11 in wintertime and 0.41 in summertime. Fig. 6 shows the
simulated GrIS-averaged SMB and the NAO index for NPS-0k in winter and summer. Like in MAR, in winter, the
correlation coefficient is slightly negative (-0.22), meaning that a positive NAO index is preferably (but not systematically)
associated with SMBs lower than average on Greenland (or net accumulation, since there is virtually no melting in winter,
see Table 2). In summer, SMB anomalies are more strongly correlated with the NAO index (0.62) than in the MAR
simulation. Note however that the correlation estimate from MAR and ERA-Interim is probably not robust due to the short
time series used (34 years). In order to test the stability of this correlation, we splitted up the NAO index and SMB time
series simulated by NPS-0k into 8 consecutive chunks of 34 years and found that the correlation coefficients for these 34-
year time-spans ranged from 0.42 to 0.71, which is compatible with the result obtained from MAR and ERA-Interim.

### 4 Greenland climate, NAO and GrIS SMB during the 130 ka, 115 ka and preindustrial periods

### 4.1 Changes in solar radiation and climate response

The orbital eccentricity, precession and obliquity modulate the solar flux at the top of the Earth's atmosphere. The
eccentricity is the deviation of the orbit from a perfect circle and is the only orbital parameter that can modify the global
year-mean solar irradiance per unit surface area. The precession is the change in the orientation of the Earth's rotational axis
and the obliquity is the angle between the Earth's rotational axis and its orbital axis. Both parameters alter the repartition of
solar energy by latitude bands. The eccentricity and precession parameters mainly modulate the Earth-Sun distance, whereas
obliquity mainly determines the latitude with largest solar irradiance. During Eemian, changes in precession led to
significant insolation changes due to the high eccentricity. On top of that, high (low) obliquity is associated with less (more)
insolation at middle and high latitudes. Hence, since the obliquity increases with time from the beginning (130 ka) to the end
(115 ka) of the interglacial period, high latitudes received less irradiation at 115 ka than at 130 ka.



Zonal averages of monthly and annual insolation anomalies between the Eemian and the preindustrial periods are shown in

Fig. 7. The 130 ka is characterized by positive annual anomalies at high latitudes with very different seasonal cycles between the Northern Hemisphere (NH) and the Southern Hemisphere (SH). Strong positive anomalies (>50 W/m2) prevail north of 20°N during approximately two months (April-May) whereas in the South Hemisphere positive anomalies only appear south of 60°S during approximately one month. In tropical regions, anomalies are negative for six consecutive months and the annual mean insolation anomaly is close to zero. At 115 ka, insolation anomalies are broadly with opposite sign compared to

those of 130 ka. More (less) solar energy reaches tropical (polar) regions for 115 ka than for 130 ka. The monthly gradient of insolation anomaly during spring and autumn decreases between 130 ka and 115 ka. The Earth's orbital parameters lead to zonal annual changes of insolation from the preindustrial period that depend on the seasonality of solar radiation. In the Arctic, the annual increase (decrease) of insolation anomalies at 130 ka (115 ka) compared to the preindustrial results in a warmer (cooler) climate from March to June (April to July). In order to document the near-surface response to the changes in

insolation, the simulated NPS-130k and NPS-115k 2m-temperature summertime anomalies (with reference to NPS-0k simulation) are plotted in Fig. 8. The three NPS experiments only differ by the orbital parameters and therefore changes in mean states and variabilty can be attributed to differences in solar forcing. In NPS-130k, the largest positive 2m-temperature anomalies, as high as 4 °C, appear in the central part of the GrIS (Fig. 8a), where the high elevation leads to cold and dry conditions. This anomaly suggests that in this region the ice sheet and atmosphere interact through a thermodynamic balance.

In this region, the mean circulation is mostly controlled by local processes. Conversely, in NPS-115ka, the largest cooling anomalies do not correspond with the highest elevations, suggesting that even in the central part of the GrIS, the mean climate is mainly determined by atmospheric dynamics rather than local processes. The largest negative temperature anomalies occur in the northern part of central Greenland (Fig. 8b), which are influenced by cold northerly winds blowing from the ice-covered Arctic Ocean to Greenland, cooling the near-surface atmosphere.

We finally examine changes in the seasonal (DJF and JJA) means of accumulation, melting and SMB averaged over the Greenland mask for interglacial and preindustrial climates (Table 2). Under all climates, there is more accumulation in summertime than in wintertime and, as expected since accumulation increases with temperature, the accumulation is stronger at 130 ka for both seasons. Even if simulated melting rates are underestimated, since the same model is used for all experiments, we compare them in terms of relative values. As a consequence of obliquity changes, melting is much larger at

130 ka than for preindustrial and 115 ka.

The spatial structure of the JJA NAO patterns does not depend much on the considered period, as shown in Fig. 9. The southern positive node extends farther west and south during 130 ka, and the extension of the northern negative node is smaller at 130 ka than at 115 ka and preindustrial. The total variance of SLP explained by the EOF1 does not depend much either on the period, and is equal to 30.3 % , 33.0 % and 29.8 % respectively in NPS-0k, NPS-115k and NPS-130k.

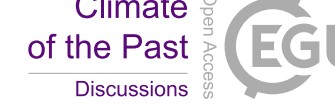



## 4.2 The link between NAO and SMB

Correlation maps between SMB and the NAO index for preindustrial and both interglacial climates are plotted in Fig. 10. In winter (Figs. 10a-c), the overall similarity in the patterns of SMB shows that even if the representation of the GrIS orography in CNRM-CM is rather coarse, the model reasonably simulates the rising of moisture in the atmosphere over the steepest parts of the GrIS. Also, we note that variations in zonal annual insolation do not significantly modulate accumulation. The correlation maps show that the link between the NAO index and SMB is relatively strong under all climates in the northwestern (negative correlations) and northeastern (positive correlations) parts of the central GrIS. No melting occurs in winter and the SMB reduces to net accumulation. The strongest negative correlations in the northwestern part of central Greenland are due to the lower snow precipitation during the negative phase of the NAO, contrasting with strong westerly advection of moisture in South Greenland.

In summer (Figs. 10d-f), patterns of SMB are different for each climate and correlation coefficients are positive, increasing from northeastern central Greenland (dry zone) towards the coast. At 130 ka (Fig. 10f), the large mid-to-high latitude warming explains the relatively wide area with negative SMB along the GrIS margins. In particular, strong snow melting occurs in the northeastern part of the GrIS, suggesting this area to be a vulnerable part of the GrIS under a warm climate since melting is not compensated by accumulation. Similarly, Born and Nisancioglu (2012) concluded that at 126 ka, the strongly negative annual mean SMB in the northeastern part of the GrIS leads to a significant thinning of the ice-sheet, which is amplified by the ice elevation feedback. At 115 ka (Fig. 10e), in the southern part of the GrIS, positive and negative SMB are strongly correlated with NAO. On the margins the cold summer climate inhibits melting, whereas at higher elevations, the SMB becomes positive as a response to increased accumulation due to atmospheric moisture transport. Note that at 130 ka, the northeastern part of the GrIS shows slightly negative SMB not correlated to the NAO index. The preindustrial SMB patterns (Fig. 10d) are intermediate between 115 ka (Fig. 10e) and 130 ka (Fig. 10f). In North-West (South) Greenland, they are rather similar to those of 130 ka (115 ka). Over most of Greenland the preindustrial patterns of correlation are rather similar to 130 ka despite lower values in particular along coastal areas. In the northeastern part of the GrIS (up to 70°N), preindustrial SMB is less correlated to the NAO index than at 130 ka. Under preindustrial climate, the strongest correlations are confined to the southern margins, as well as at 115 ka, reflecting reduced snow precipitation at lower elevations during the negative phase of the NAO, when the atmospheric flow is less pronounced. More inland, the strong correlations are due to the barrier effect of the GrIS that forces the rise of atmospheric moisture and subsequent snow precipitation (Bromwich et al., 1999; Folland et al., 2009; Fettweis et al., 2011; Auger et al., 2017).

To better understand these summertime differences in SMB and NAO correlations for the contrasted climates of the Eemian, the analysis now focuses on the influence of NAO phases on accumulation and melting.



### 4.3 Impact of the summertime phases of the NAO on SMB and its components

As was done for NPS-0k (Fig. 6), the correlation between the GrIS-averaged summertime SMB anomalies and the NAO index was plotted for both interglacial climates (Fig. 11). The correlation coefficients for NPS-0k, NPS-115k and NPS-130k are respectively equal to 0.61, 0.62. and 0.56. In order to go further in the analysis, we sampled positive and negative phases of the NAO and computed grid-point correlation maps with accumulation (Fig. 11) and melting (Fig. 12). Summers with an absolute value of the NAO index less than one standard deviation were excluded. The statistical significance of correlation coefficients is estimated at the 99% confidence level.

In the eastern part of the GrIS, significant correlations between accumulation and NAO only occur for positive NAO phases. In North Greenland, significant correlations are only seen under 130 ka climate for positive NAO phases. The link between accumulation and NAO is rather strong in South Greenland with correlations greater than 0.25. The positive phase of the NAO favours accumulation in most of South Greenland in preindustrial (Fig. 12a) and 130 ka (Fig. 12c), i.e. under warm climates, whereas under the colder 115 ka climate, the negative phase of the NAO favours accumulation (Fig. 12e). The accumulated precipitation primarily arises from oceanic evaporation and atmospheric transport towards South Greenland. Oceanic evaporation is related to surface atmospheric forcing and SST anomalies which can be generated by NAO phases. For example, the negative phase of the NAO is associated with negative SST anomalies from Baffin Bay to the Greenland Sea and positive SST anomalies in the central North Atlantic (Pinto and Raible, 2012 and references there in). More over, at 115 ka, the latitudinal insolation gradient (Fig. 7, bottom) induces a larger northward atmospheric moisture transport from the warmer tropical ocean, supplying higher latitudes with moisture (Ramstein et al., 2005). Finally, central Greenland sees less precipitation, since moist air masses tend to generate precipitation, hence getting drier on their way towards inland Greenland. More over, the presence of the anticyclone over the GrIS induced by downdraft air masses over the cold northern ice sheet also prevents the penetration of moist air masses into the interior of Greenland (Merz et al., 2014 and references there in).

For the positive phase of the NAO, the correlation of melting with the NAO index is greater than 0.25 mainly along the steepest margins of the GrIS except along the eastern coast north of 70 °N for all climates (Figs. 13a-c). For the negative phase of the NAO, melting tends to be correlated with the NAO index only for 130 ka, along the eastern coast north of 70 °N (Figs. 13d-f). Note that the preindustrial displays no significant link between the negative phase of the NAO and melting.

### Conclusions and perspectives

In this paper we examined the link between the Surface Mass Balance of the Greenland Ice Sheet and the North Atlantic Oscillation (NAO) for the last interglacial and preindustrial climates. For this study we developed a configuration of CNRM-CM5.2 with enhanced atmospheric horizontal resolution on Greenland (40 to 55 km), which is reasonably suited for simulating the spatial variability of accumulation and surface melting. On the basis of a comparison with a regional





simulation performed with MAR for 1979-2012, we showed that the simulated accumulation in our preindustrial simulation is realistic, whereas surface melting is much underestimated due to the too high minimum albedo (0.8) used in CNRM-CM5.2.

For all climates, the anomalies of the averaged SMB over all of Greenland and the NAO index (normalized leading PC of detrended SLP anomalies) are weakly correlated in wintertime (around -0.2) and strongly correlated in summertime (around 0.6). These correlations are in broad agreement with those between SMB simulated by MAR and the NAO index computed from ERA-Interim for the period 1979-2012. Note that, on the one hand, our estimate of the summer correlation computed from MAR and ERA-Interim is probably not robust due to the relatively short sampling time (34 years). In summer, the underestimation of surface melt does not seem to affect much the correlation between the SMB averaged over all of Greenland and the NAO index because it remains consistent with results of Fettweis et al. (2013) highlighting the link between surface melting over the GrIS and the negative phase of the NAO.

This study also emphasized the spatial pattern of the link between SMB and the NAO index. In winter, this spatial pattern is similar for all mean states, with negative (positive) correlations in the western (eastern) part of central Greenland. Both regions are characterized by relatively dry conditions in contrast with South Greenland. More over, the similarity between regional patterns of the SMB as well as its correlation with the NAO index under all climates indicate that the variability of insolation has a weak influence on the patterns of accumulation in winter. In summer, the spatial patterns of SMB and its correlation with the NAO index depend on the climate. The link between NAO and SMB is all the stronger as the climate is warm (e.g. stronger at 130 ka than at 115 ka) and gets stronger from the northwestern part of central Greenland to the margins. Even if our model strongly underestimates melting compared to MAR, strong negative SMB appears, especially in the northeastern part of Greenland under warm climates (preindustrial and 130 ka). This result shows that in a warm climate, the northern and northeastern parts of the GrIS could be nibbled. In South Greenland, very strong SMB gradients confined to the margins are associated with strong correlations with NAO under preindustrial and 115 ka climates. More inland, the SMB is positive because the southwesterly and southeasterly flows strongly interact with the topography. In South Greenland, the simulated preindustrial SMB is similar to SMB at 115 ka, whereas in North Greenland it can be viewed as intermediate between 115 ka and 130 ka patterns.

The last part of this work highlights the influence of both positive and negative phases of the NAO on accumulation and melting. Accumulation in South Greenland varies preferentially with the positive (negative) phase of the NAO in a warm (cold) climate. Under warm climates, the positive phase of the NAO favours the large scale advection of moisture in South-West Greenland and subsequent precipitation. At 115 ka, the accumulation tends to be controled by the negative phase of the NAO. In summertime, the negative phase of the NAO is significantly correlated to melting only in North-East Greenland at 130 ka, whereas its positive phase promotes melting along the margins of the GrIS under all climates. The representation of the spatial structures of accumulation and melting and their links with NAO are both significantly improved due to the enhancement of horizontal resolution on Greenland in the NPS configuration compared with the CNRM-CM5 configuration



used for CMIP5. Future work will investigate the critical influence of sea-ice and SST to establish the robustness of the
simulated links between SMB and its components with both phases of the NAO.

**Acknowledgements**

The authors would like to thank X. Fettweis (University of Liège, Belgium) for providing a regional simulation from MAR
for the period 1979-2012. The main author is grateful to Aurore Voldoire for her help in handling the CNRM-CM model.
Supercomputing resources were provided by Météo-France/DSI that we thank for their support and guidance. The figures
were produced thanks to the NCAR Command Language (NCL) Software (doi:10.5065/D6WD3XH5NCL) and the PCMDI
Climate Data Analysis Tools (CDAT), the use of which is hereby acknowledged.

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



**Table 1. Astronomical forcing for all simulations after Berger (1978) in degrees.**

**\* The precession is the longitude of the perihelion relative to the moving vernal equinox minus 180°.**

| simulation | Period | Eccentricity | Precession* | Obliquity |
|---|---|---|---|---|
| NPS-0k | preindustrial | 0.01672 | 102.0 | 23.446 |
| NPS-115k | 115 ka BP | 0.04142 | 110.9 | 22.405 |
| NPS-130k | 130 ka BP | 0.03821 | 228.3 | 24.242 |





**Table 2. Winter (DJF), summer (JJA) and annual mean accumulation, melting and SMB averaged on the GrIS for
the MAR (1979-2012) simulation and for the NPS simulations at preindustrial, 115 ka and 130 ka (years 1-280). Units
are in m/yr WE.**

| Period | Accumulation | | Melting | | SMB | |
|---|---|---|---|---|---|---|
| | DJF | JJA | DJF | JJA | DJF | JJA |
| **MAR** | 0.319 | 0.343 | 0.000 | -1.683 | 0.318 | -1.343 |
| **PI** | 0.238 | 0.241 | 0.000 | -0.349 | 0.238 | -0.108 |
| **115 ka BP** | 0.225 | 0.235 | 0.000 | -0.206 | 0.225 | 0.029 |
| **130 ka BP** | 0.241 | 0.276 | 0.000 | -0.901 | 0.241 | -0.626 |




**Figure 1: 2m temperature biases between the mean states from both preindustrial simulations (top) NPS-0k and (bottom) CMIP5**
**relative to the ECMWF reanalysis ERA-Interim (years 1979-2012) in (left) boreal winter (DJF) and (right) summer (JJA).**



**Figure 2: Annual mean precipitation (top) and monthly mean precipitation for January (middle) and July (bottom) in the preindustrial simulations (years 1-280) NPS-0k (left) and CMIP5 (middle) and ERA-Interim (1979-2012) (right).**





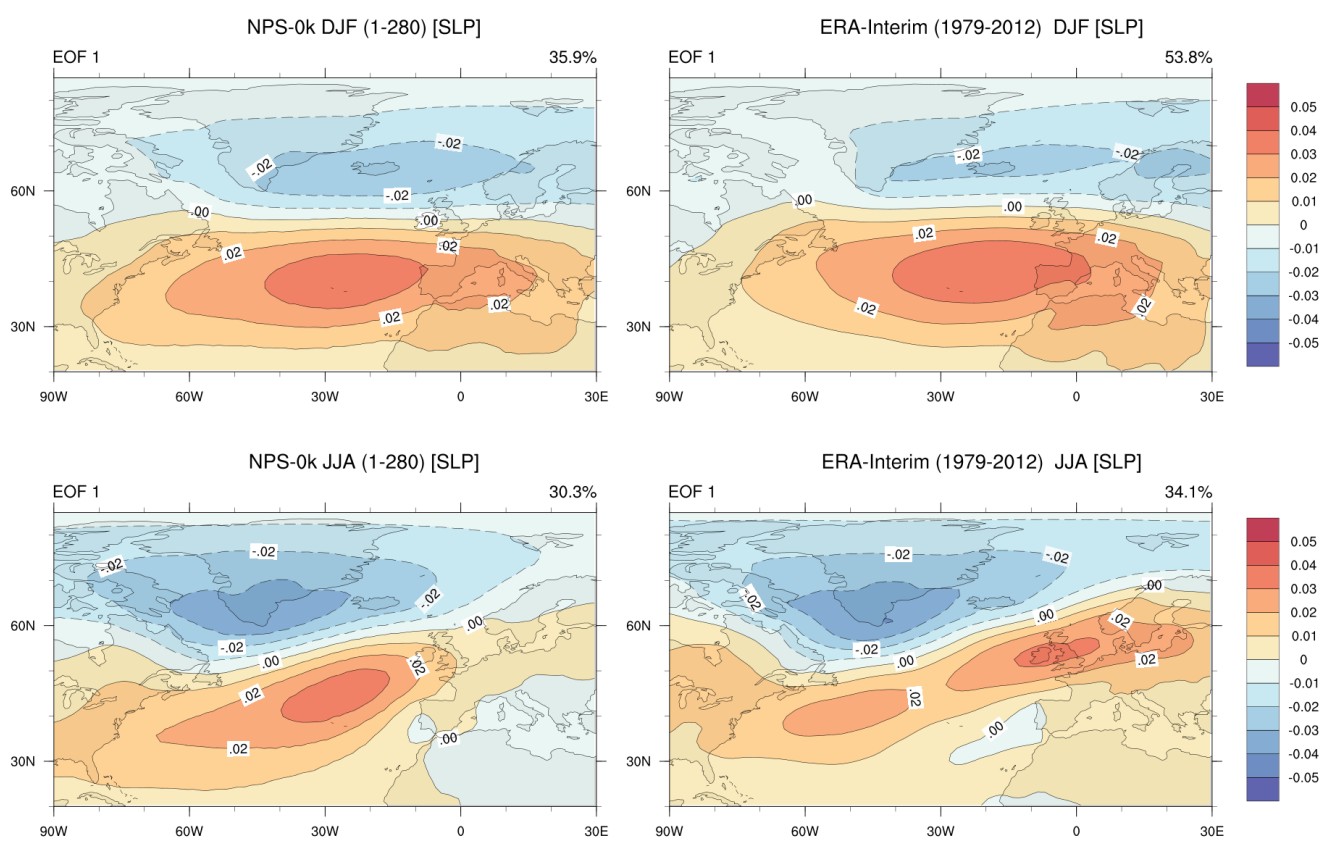

**Figure 3: Leading EOF of SLP for (left) NPS-0k (years 1-280) and (right) ERA-Interim (years 1979-2012) in winter (DJF, top row) and in summer (JJA, bottom row).**





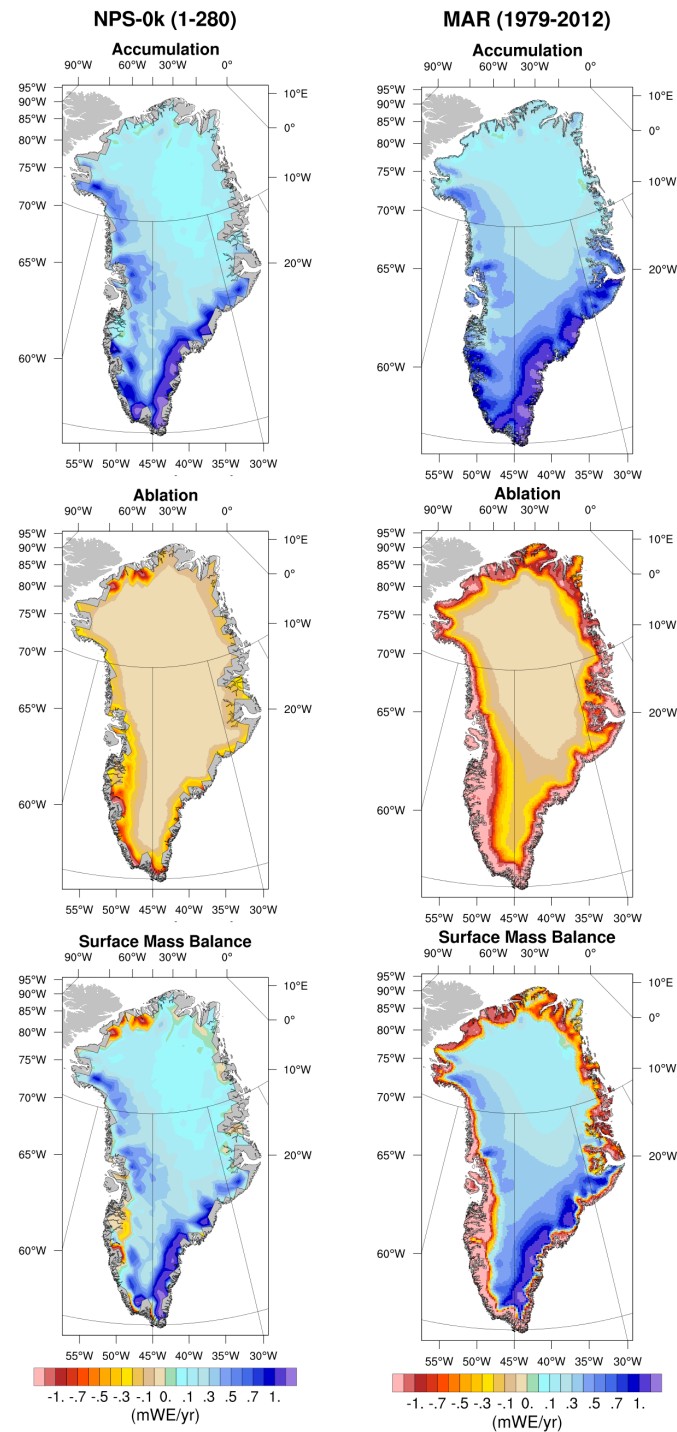

**Figure 4: Annual mean accumulation (top), ablation (middle) and SMB (bottom) over Greenland from NPS-0k (left) and MAR (right) .**




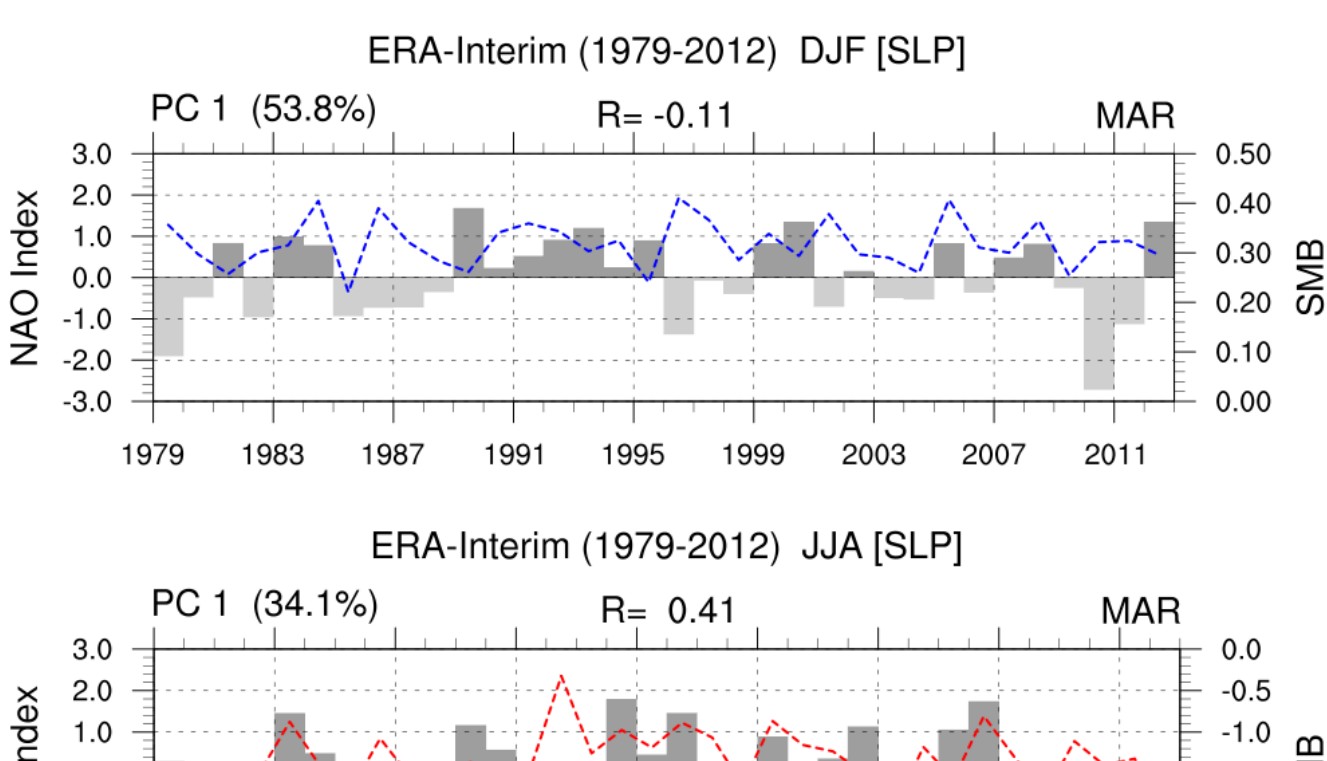

**Figure 5: (grey bars)** Time series of normalized leading PC derived from an EOF analysis of ERA-Interim SLP over the extra-tropical North Atlantic (20N-70N;90W-40E) and **(dashed lines)** average SMB over Greenland simulated by MAR for winter (DJF, top) and summer (JJA, bottom).





**Figure 6: (grey bars) Time series of normalized leading PC derived from an EOF analysis of SLP over the extra-tropical North Atlantic and (dashed lines) average SMB over Greenland in NPS-0k for winter (DJF, top) and summer (JJA, bottom).**





**Figure 7: Zonal mean departures of (left) monthly and (right) annual insolation from preindustrial conditions for (top) 130 ka and (bottom) 115 ka BP. On the left panels, contours are drawn every 5 W m-2 from -50 to 50 W m-2. Full lines are for positive deviations (Eemian values larger than preindustrial) .**





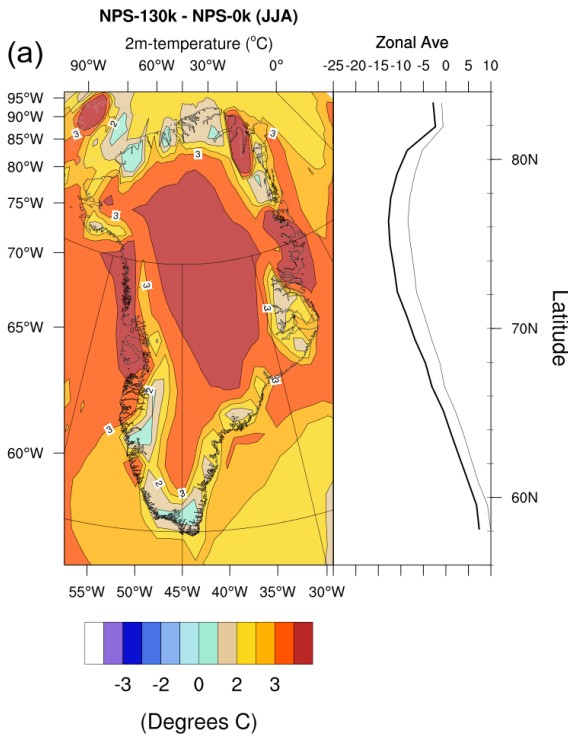
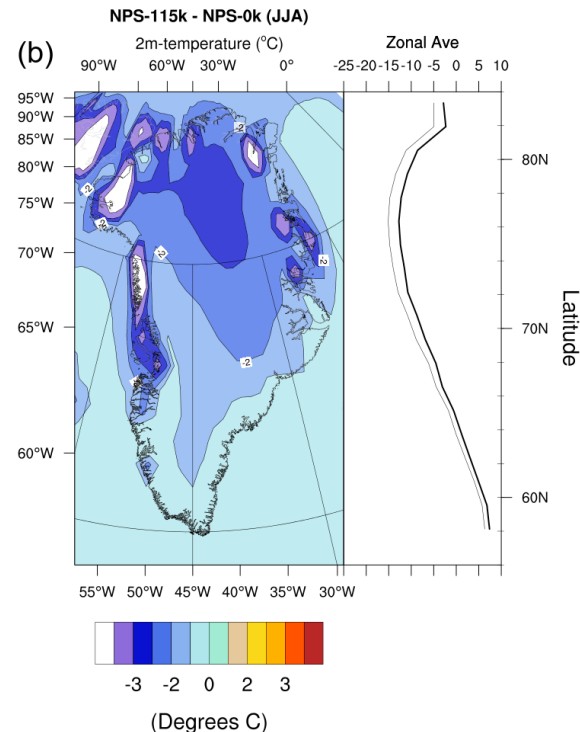

**Figure 8: Summer (JJA) 2m temperature anomalies for 130 ka (left) and 115 ka (right) from preindustrial conditions. In both**
**plots, the left and right subplots respectively represent the spatial pattern of anomalies and their zonal average.**






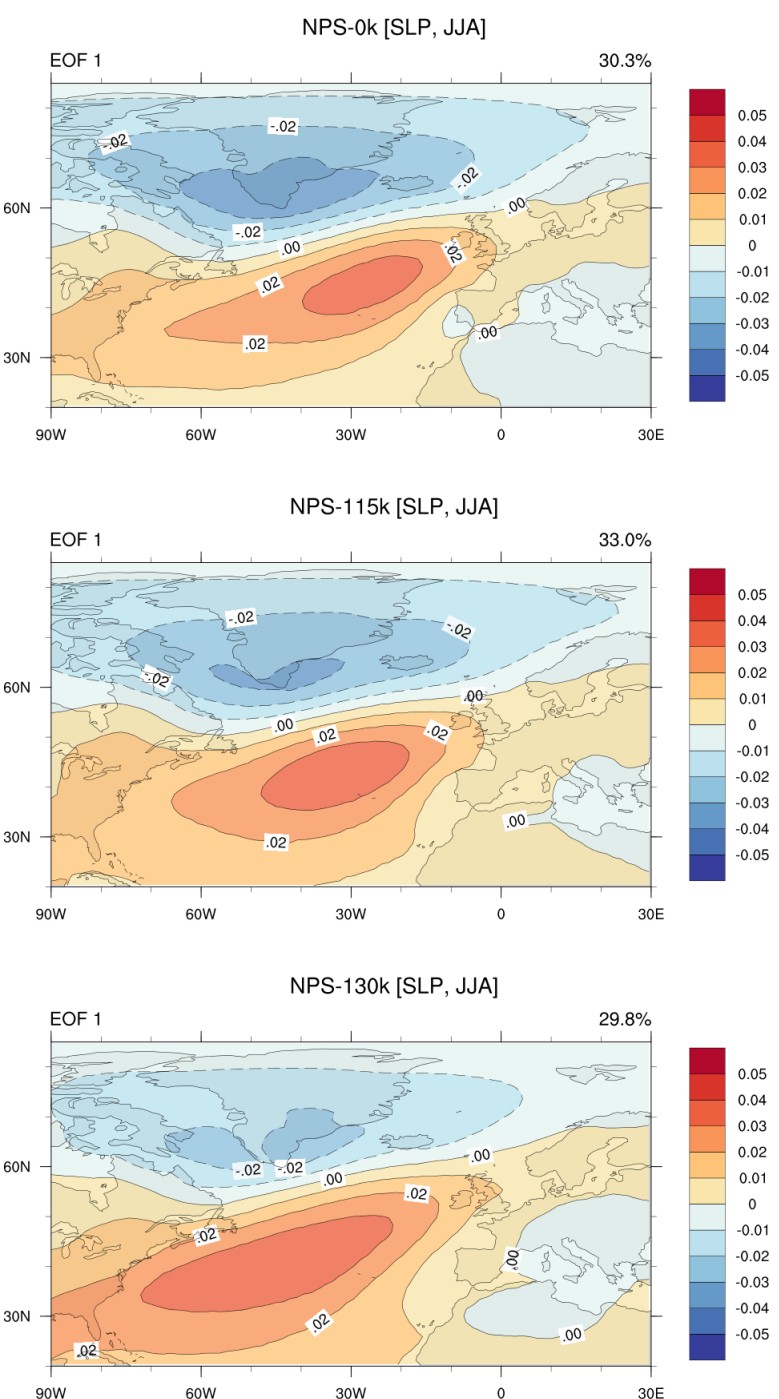

**Figure 9: Summertime (JJA) leading EOF of SLP for (top) NPS-0k, (middle) NPS-115k and (bottom) NPS-130k (years 1-280).**



**Figure 10: SMB (in m WE/yr) and (superimposed) the correlation between SMB and the NAO index for (left) preindustrial, (middle) 130 ka and (right) 115 ka in (top) winter and (bottom) summer.**

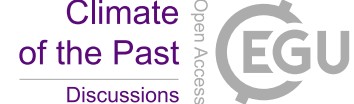

**Figure 11: (grey bars)** Time series of normalized leading PC derived from an EOF analysis of SLP over the extra-tropical North Atlantic and **(dashed lines)** average SMB over Greenland in NPS-115k **(top)** and NPS-130k **(bottom)** for summer (JJA).

**Figure 12: Spatial correlation between accumulation and the NAO index for (left) preindustrial, (middle) 115 ka and (right) 130 ka. Top (bottom) row: positive (negative) NAO situations (sampled for NAO indices with absolute value higher than one stantard deviation). Dotted areas represent correlations significant at the 99% level.**





**Figure 13: Same as Fig. 12, but for spatial correlation between melting and the NAO index.**