# Peer review of "Link between the North Atlantic Oscillation and the Surface Mass Balance components of the Greenland Ice Sheet under preindustrial and last interglacial climates: a study with a Coupled Global Circulation Model"

_Climate of the Past, 2018_

## Referee Comment (RC1) · Anonymous Referee #1 · 2 May 2018

This paper shows that the link between NAO and SMB over Greenland has been changed through the centuries by using the CNRM-CM5.2 global model. The paper is interesting to read and deserves to be published in CP. However, some (major) improvements are still needed before publication.

Major:

1. In Fettweis (2007), seasonal 2D correlations with NAO was shown (see Fig 15 and

[Figure]

Fig16). Such similar figures should be shown with the MAR data used here and the CNRM based present climate reconstruction to check if CNRM is able to simulate the current pattern of NAO impacts on SMB. The validation by using the R value (correlation) shown in Fig 5 and Fig 6 is not enough for me. A 2D validation is needed here as the paper discusses 2D changes in the correlation with NAO.

2. The ERA forced 1979-2012 period is used here as validation for pre-industrial climate but since the end of the 1990's, we have observed a switch of NAO toward negative value in summer as remembered in this manuscript. This issue should be discussed or the period 1980-1999 should be used as validation. This reference period 1980-1999 was selected in Fettweis et al (2013b) for this reason because surface melt GrIS records were observed over the 2007-2012 period (included in the reference period used here) which is not representative of the present or pre-industrial climate.

Minor:

1. Fig1: What is the interest of showing the whole globe while only the North Atlantic area is discussed here ? A zoom over the area of interest will be more useful.

2. Fig2: as only positive values are shown, the legend could be adapted.

3. As said earlier, what is the interest of showing Fig 5, fig 6 and Fig 11. Only the statistics listed here are useful for me and can be put in a table. Are there any trends in the CNRM based time series? Over 1979-2012, the MAR based SMB should significantly decrease as well as the JJA NAO index.

4. Why Fig 12 and Fig 13 are black and white and not in colour? Why only the correlation with accumulation is show over summer in Fig 12? Over these figures, it is difficult to distinguish which is significant or not.

5. Section 4.3 : Fig 11 and Fig 12 are referenced in the text instead of Fig 12 and Fig 13 (ex: line 284).

---

## Referee Comment (RC2) · Anonymous Referee #2 · 8 May 2018

The manuscript studies the connection between the NAO index and SMB of the Greenland ice sheet using a set of experiments with an AOGCM for different orbital configurations. The model uses a configuration of increased spatial resolution over the region of interest, which improves the representation of some atmospheric circulation features compared to the standard CMIP5 configuration. Correlation analysis reveals spatial and temporal patterns of correlation between the NOA and SMB. Despite improved resolution, the representation of surface melt is poorly represented in the model. With

this, I feel there is limited confidence that the model is the right tool to study the NAO-*SMB* relationship. I have made two suggestions in the general comments below how I see the study may be modified to circumvent this problem, both requiring a substantial reworking of the material, i.e. major revisions.

— General comments —

The model does not perform well in simulating surface melt and runoff, which is an important component to the surface mass balance at present, it will become more important in the future and likely was important during the Eemian warm period. This implies an important caveat for interpreting links between the NAO and SMB as put forward in the manuscript. As it stands now, the shortcomings of the model in terms of melting are also not well presented, with contradicting statements (see specific comments below). I was wondering if the authors could focus on precipitation changes (instead of SMB) and their relation to the NAO as a more robust feature of the model. Another possibility may be to look at a precursor of melt, like the 700 hPa temperature, which appears to be a good predictor for surface melt according to Fettweis et al. (2013a).

While it is recognised in the manuscript that earlier research has shown that "changes in atmospheric circulation" are responsible for a large part of the summer warming in Greenland (citing Fettweis et al., 2013), an important distinction put forward by Hanna et al. (2013) is not further discussed: they find that "Greenland coastal summer temperatures and Greenland Ice Sheet (GrIS) runoff since the 1970s are more strongly correlated with the Greenland Blocking Index (GBI) than with the NAO Index". In the context of the present paper concerned exactly with the relation between atmospheric circulation and GrIS SMB, it seems in place to also discuss the Greenland Blocking Index. Possibly the model in this study does not represent the GBI nor the relation to Greenland SMB very well. In that case, this should be clearly presented and discussed as another limitation of the model.

The correlation analysis is an important element of the manuscript and reveals important spatio-temporal differences in the relationship between NOA and the Greenland SMB (or at least precipitation, see above). What I miss in the paper is a step beyond the correlation analysis to help the reader understand what this work really implies. Should we expect a stronger influence of the NAO on Greenland in the future or during the Eemian? What would that imply for a possible distribution of melt and precipitation changes? Does the seasonal difference in the relationship play an important role now and how will that change in the future?

In the introduction, the study is fully motivated with a perspective on the future. Given the different forcing mechanism between the Eemian and the future (orbital vs. GHG), one could question if the chosen experiments (130 and 115 kyr BP orbital configuration) are really a good choice to learn something about future changes. In my opinion, the future perspective could be a much less important element in this paper and more focus be placed on understanding the Eemian climate itself. While the idea to learn something about the future by looking at the past is one of the well established and accepted motivations in paleo research, the opposite perspective can also be rewarding and should probably be added for a more balanced view. The fate of the GrIS during the Eemian e.g. remains a scientific problem of high relevance, which could be mentioned and discussed.

— Specific comments —

L22: Distinguish between Fettweis et al., 2013a and 2013b in the manuscript.

L42: Could you please clarify the term "surface temperature feedback". Often a feedback is named mentioning two components that have mutual dependencies like SMB-surface elevation feedback or surface temperature - albedo feedback.

L61: "Better the link between NAO *variations* and ..."

L62: The terminology "warm and cool phase of the Eemian" may not be correct. I

would refer to the studied time slices as "the warm climate of the Eemian" and "the cold climate of the penultimate glacial inception" or similar.

L67: The MAR model could be introduced much earlier, e.g. when discussing results of Fettweis et 2013 (L35ff).

L78-81: This is a confusing description. As long as there is no coupling to an ice sheet model, it is standard for an AOGCM to operate with a fixed surface topography over land. As far as I can see it, this has nothing to do with technical requirements of the snow pack model as described here. It would be interesting to describe instead if and how the snow-pack model differs from other GCMs and from the MAR model, which I suspect has a full physical solution to the problems you are describing.

L81-83: Since there is no ice-dynamical process in this model at all, it seems strange to evoke the idea of a calving flux. I think it would be far simpler to say that all precipitation over ice-covered land is equally distributed over the ocean north of 60N, while the snow pack evolution is calculated diagnostically, without contribution to the mass budget. It should be clarified that the instantaneous relocation of this mass (freshwater?) as an additional forcing does not have any influence on the ocean response.

L86: A resolution of 40-50 km is still relatively low compared to the resolution of state-of-the-art regional climate models (MAR at 15km, RACMO at 11km). This should mentioned here.

L120: If model bias and climate change signal are combined, how do we tell them apart? Is there maybe another experiment that could separate these two factors?

L124ff: Is this discussion really important for the GrIS? Consider discussing the biases for Greenland in more detail instead.

L169: Is it elevation or surface slope that has an important impact on precipitation amounts? Clarify.

L178: Clarify if this masking includes ice caps and glaciers in the periphery of the

Greenland ice sheet.

L185: You attribute most of the underestimation of melt to the albedo limit. Why is that limit in place? Are there other shortcomings of the snowpack model worth mentioning? How does the snowpack model compare in complexity and included processes to the one in MAR? If resolution is an important limitation, how does the model compare to low resolution versions of MAR (Franco et al., 2012).

L194: Could add a few references after "Greenland" as a reminder.

L191: I strongly disagree with this statement. The model is clearly not reproducing the melting well and therefore shows considerable shortcomings to represent the SMB. The statement is in clear contradiction to the description L184 and L312.

L195: Could you please clarify if the NAO index is here calculated based on the normalised PC as described at line L156? In other words, is the NAO index definition the same for the ERA-based correlation with MAR SMB as the CNRM-CM5.2 correlation with CNRM-CM5.2 SMB?

L215: Are "changes in precession" meant compared to pre-industrial or to other times during the Eemian?

L246: Could you find a better word instead of "node"? This is the first time this term is used. Maybe 'region'?

L310: Again, I think this statement may be true for accumulation, but clearly not for melting.

L317: There is "another hand" missing in this sentence or somewhere in the following.

L331: Not sure what "nibbled" means, please revise. Interesting to speculate on the impact of the Greenland ice sheet during the Eemian, extend if possible.

L344: This final statement may raise the suspicion that the findings in this paper are not yet established to be robust and may be subject to change. Maybe just a question

of formulation. Revise.

Figure 2 Precipitation is defined positive. Maybe adjust the colour scale accordingly?

Figure 4 This figure clearly shows that ablation and SMB are very poorly represented in NPS-0k.

Can you show the sublimation E subtracted from P to get accumulation C in the top panel (maybe as a supplement)? It seems to have a large impact on the resulting C. It also seems to have large spatial variability. Is that expected?

— References —

Franco, B., Fettweis, X., Lang, C., & Erpicum, M. (2012). Impact of spatial resolution on the modelling of the Greenland ice sheet surface mass balance between 1990–2010, using the regional climate model MAR. The Cryosphere, 6(3), 695–711. http://doi.org/10.5194/tc-6-695-2012

---

## Author Comment (AC1) · 21 Jun 2018

Interactive comment on: "Link between the Surface Mass Balance of the Greenland Ice Sheet and the North Atlantic Oscillation under preindustrial and last interglacial climates: a study with a Coupled Global Circulation Model" by S. Ramos Buarque and D. Salas y Melia Anonymous Referee #1

This paper shows that the link between NAO and SMB over Greenland has been

changed through the centuries by using the CNRM-CM5.2 global model. The paper is interesting to read and deserves to be published in CP. However, some (major) improvements are still needed before publication.

Major: 1. In Fettweis (2007), seasonal 2D correlations with NAO was shown (see Fig 15 and Fig16). Such similar figures should be shown with the MAR data used here and the CNRM based present climate reconstruction to check if CNRM is able to simulate the current pattern of NAO impacts on SMB. The validation by using the R value (correlation) shown in Fig 5 and Fig 6 is not enough for me. A 2D validation is needed here as the paper discusses 2D changes in the correlation with NAO.

In the new version of the paper, Fig 5 and Fig 6 have been removed. We agree that the added-value of these plots was low and decided to provide the correlation coefficients only in the text of the paper. Instead, we have inserted figures showing the seasonal spatial correlation of accumulation, ablation and surface mass balance with the NAO index. By contrast with the paper by Fettweis (2007), we chose not to display correlations for the intermediate seasons (MAM and SON), in order to focus on DJF and JJA, like in the rest of the paper. In this answer, you will find hereafter (FYI) the correlation between precipitation and the NAO index for all seasons.

2. The ERA forced 1979-2012 period is used here as validation for pre-industrial climate but since the end of the 1990's, we have observed a switch of NAO toward negative value in summer as remembered in this manuscript. This issue should be discussed or the period 1980-1999 should be used as validation. This reference period 1980-1999 was selected in Fettweis et al (2013b) for this reason because surface melt GrIS records were observed over the 2007-2012 period (included in the reference period used here) which is not representative of the present or pre-industrial climate.

In the new version of the paper, we now systematically use the 1980-1999 time-span for validation, instead of 1979-2012 previously, to be more consistent with preindustrial conditions. Among other results, this change in validation period affects the correlation

we provide between SMB and NAO indices for MAR.

Minor: 1. Fig1: What is the interest of showing the whole globe while only the North Atlantic area is discussed here ? A zoom over the area of interest will be more useful. We wanted to show the whole globe for a general, view of model biases. Following the reviewer's recommendation, we changed the domain of Fig. 1 to represent only the Arctic and the North Atlantic. However, since our model is a global one, we chose not to restrict the figure to Greenland, in order to place the biases over the GrIS in a wider, still not global context.

2. Fig2: as only positive values are shown, the legend could be adapted. We adapted the legend to follow this recommendation.

3. As said earlier, what is the interest of showing Fig 5, fig 6 and Fig 11. Only the statis- tics listed here are useful for me and can be put in a table. We agree with these comments, and removed Fig. 5, 6 and 11. The table hereafter shows correlations between the NAO index and the GrIS-averaged accumulation, melting and SMB for MAR and NPS under all climates. However, we have chosen not to integrate it into the paper since we now focus more on the 2D-correlations.

Are there any trends in the CNRM based time series? The trends are very small in the time series provided by CNRM-CM. However, we removed the trend for plotting the time series and computing the correlations, just as we did for MAR due to the large SMB trend over 1979-2012. Over 1979-2012, the MAR based SMB should significantly decrease as well as the JJA NAO index. The MAR based SMB and NAO time series were detrended, in order to correlate just interannual variations, not the trends over 1979-2012.

4. Why Fig 12 and Fig 13 are black and white and not in colour? New figures were plotted with color shading. Why only the correlation with accumulation is show over summer in Fig 12? We added the correlation of winter accumulation with NAO+ and NAO- (new Fig 11). Note that for DJF, significant parts of the GrIS show negative

correlations (unlike for JJA). Hence we adapted the range of plotted values accordingly for Figs. 11, 12 and 13. We adapted the text accordingly (see Sec. 4.2 and 4.3) Over these figures, it is difficult to distinguish which is significant or not. The dashed areas corresponding to significant correlations are now easier to see thanks to the colored background.

5. Section 4.3 : Fig 11 and Fig 12 are referenced in the text instead of Fig 12 and Fig 13 (ex: line 284). Thanks for this comment, done.
* * *
[Figure]

[Figure]

[Figure]

**Fig. 1.**

|  | Accumulation | | Melting | | SMB | |
|---|---|---|---|---|---|---|
|  | DJF | JJA | DJF | JJA | DJF | JJA |
| **MAR** | **-0.21** | **0.54** | **-** | **0.37** | **-0.21** | **0.40** |
| NPS-0k | -0.22 | 0.48 | - | 0.51 | -0.22 | 0.62 |
| NPS-115k | -0.11 | 0.43 | - | 0.56 | -0.11 | 0.62 |
| NPS-130k | -0.04 | 0.48 | - | 0.43 | -0.04 | 0.56 |

Seasonal (DJF and JJA) correlations between accumulation, melting and SMB averaged on GrIS and the NAO index.

**Fig. 2.**

---

## Author Comment (AC2) · 21 Jun 2018

Interactive comment on: "Link between the Surface Mass Balance of the Greenland Ice Sheet and the North Atlantic Oscillation under preindustrial and last interglacial climates: a study with a Coupled Global Circulation Model" by S. Ramos Buarque and D. Salas y Melia

Anonymous Referee #2

[Figure]

The manuscript studies the connection between the NAO index and SMB of the Greenland ice sheet using a set of experiments with an AOGCM for different orbital configurations. The model uses a configuration of increased spatial resolution over the region of interest, which improves the representation of some atmospheric circulation features compared to the standard CMIP5 configuration. Correlation analysis reveals spatial and temporal patterns of correlation between the NAO and SMB. Despite improved resolution, the representation of surface melt is poorly represented in the model. With this, I feel there is limited confidence that the model is the right tool to study the NAO-*SMB* relationship.

* We decided to focus the paper on the relationship between NAO and the components of the GrIS SMB, rather than on the relationship between NAO and SMB itself. The title of the paper was changed accordingly to "Link between the North Atlantic Oscillation and the Surface Mass Balance components of the Greenland Ice Sheet under preindustrial and last interglacial climates: a study with a Coupled Global Circulation Model" I have made two suggestions in the general comments below how I see the study may be modified to circumvent this problem, both requiring a substantial reworking of the material, i.e. major revisions.

— General comments —

The model does not perform well in simulating surface melt and runoff, which is an important component to the surface mass balance at present, it will become more important in the future and likely was important during the Eemian warm period. This implies an important caveat for interpreting links between the NAO and SMB as put forward in the manuscript. As it stands now, the shortcomings of the model in terms of melting are also not well presented, with contradicting statements (see specific comments below). I was wondering if the authors could focus on precipitation changes (instead of SMB) and their relation to the NAO as a more robust feature of the model.

* Thanks for this suggestion. Rather than focusing on precipitation changes, we focus

on accumulation changes and their relation to the NAO, for a more direct link with SMB (except for Fig. 2)

Another possibility may be to look at a precursor of melt, like the 700 hPa temperature, which appears to be a good predictor for surface melt according to Fettweis et al. (2013a).

* We computed the correlation of melt with the 700 hPa temperature on the same domain as Fettweis et al (2013a). Compared with the correlation (0.93) reported by Fettweis et al. (2013a), we found a slightly lower correlation (0.81) with CNRM-CM5.2 over 2035-2315 (see time-series hereafter). Based on this result, even if our simulated melt is clearly underestimated, this gives confidence in assessing the relation of melt with NAO, which we do in the revised version of the paper.

While it is recognised in the manuscript that earlier research has shown that "changes in atmospheric circulation" are responsible for a large part of the summer warming in Greenland (citing Fettweis et al., 2013), an important distinction put forward by Hanna et al. (2013) is not further discussed: they find that "Greenland coastal summer temperatures and Greenland Ice Sheet (GrIS) runoff since the 1970s are more strongly correlated with the Greenland Blocking Index (GBI) than with the NAO Index". In the context of the present paper concerned exactly with the relation between atmospheric circulation and GrIS SMB, it seems in place to also discuss the Greenland Blocking Index. Possibly the model in this study does not represent the GBI nor the relation to Greenland SMB very well. In that case, this should be clearly presented and discussed as another limitation of the model.

*Indeed, the relation between atmospheric circulation indices with parameters playing a role in the SMB has been recently examined by Auger et al. (2017), who analyzed the influence of the NAO, the AMO, Icelandic Low, Azores High, regional blocking patterns, near-surface temperature and near-surface winds on precipitation in southwest Greenland. They found in particular that statistically significant correlations are higher

between precipitation and near-surface winds (0.7) than correlations between precipitation and the NAO index (0.28). More over, the relationship between GBI and other climatic indices has been examined for the period 1852–2014 by Hanna et al. (2016) who found negative and significant GBI–NAO correlations in winter. Figures hereafter show 2D correlations between melting and the NAO (left) as GBI (right) for NPS-0k (top) and MAR (bottom). The grid-point correlation map between melting / GBI is indeed very close to that of melting / NAO, however with opposite sign. That is why we have chosen not to extend the paper with discussing other atmospheric indicators.

The correlation analysis is an important element of the manuscript and reveals important spatio temporal differences in the relationship between NOA and the Greenland SMB (or at least precipitation, see above). What I miss in the paper is a step beyond the correlation analysis to help the reader understand what this work really implies. Should we expect a stronger influence of the NAO on Greenland in the future or during the Eemian? What would that imply for a possible distribution of melt and precipitation changes? Does the seasonal difference in the relationship play an important role now and how will that change in the future?

*We choose not to speculate on future climate, the best way would be to run adhoc simulations. Indeed, the aim of this study is to identify the link of NAO phases with SMB and its components in a context of natural variability, e.g. to assess if the link under current climate is robust in past climates or not. Among the implications of our work, despite the underestimated melting, the northeastern part of the GrIS seems to be vulnerable to warming at 130 ka.

In the introduction, the study is fully motivated with a perspective on the future. Given the different forcing mechanism between the Eemian and the future (orbital vs. GHG), one could question if the chosen experiments (130 and 115 kyr BP orbital configuration) are really a good choice to learn something about future changes. In my opinion, the future perspective could be a much less important element in this paper and more focus be placed on understanding the Eemian climate itself. While the idea to learn

something about the future by looking at the past is one of the well established and accepted motivations in paleo research, the opposite perspective can also be rewarding and should probably be added for a more balanced view. The fate of the GrIS during the Eemian e.g. remains a scientific problem of high relevance, which could be mentioned and discussed.

*We agree with you. Indeed, the aim of this study is to show the link of both NAO phases with mean state of accumulation, melting and SMB on interglacial and preindustrial climates. These results about past climates could serve to interpret results for future climate, but this beyond the scope of our paper.

— Specific comments —

L22: Distinguish between Fettweis et al., 2013a and 2013b in the manuscript. *Modified.

L42: Could you please clarify the term "surface temperature feedback". Often a feedback is named mentioning two components that have mutual dependencies like SMB-surface elevation feedback or surface temperature - albedo feedback. *Done. We concisely explained this feedback and added a reference.

L61: "Better the link between NAO *variations* and . . ." *Modified

L62: The terminology "warm and cool phase of the Eemian" may not be correct. I would refer to the studied time slices as "the warm climate of the Eemian" and "the cold climate of the penultimate glacial inception" or similar. Mysak (2008) defines a glacial inception as 'the transition from an interglacial to a glacial period. (. . .) the last glacial inception (LGI) at around 116 kyr BP'. Other studies, like Roche et al. (2010), define the last glacial inception as a period spanning 128-115 ka. To dismiss this ambiguity, we now refer to 115 ka as the 'late Eemian' (the definition of the end of the Eemian itself varies from 116 to 114 ka), or the cool phase of the Eemian.

L67: The MAR model could be introduced much earlier, e.g. when discussing results

of Fettweis et 2013 (L35). *The statement from this paper, that we cited is the following 'analogous atmospheric circulations in the past shows that âLij 70 % of the 1993–2012 warming at 700 hPa over Greenland has been driven by changes in the atmospheric flow frequencies.' This statement appears only in the abstract and is not related to MAR, only to reanalyses. More over, we use this result in the introduction of the paper, and, even if it was derived from a MAR simulation, we think it is too early to describe the model there.

L78-81: This is a confusing description. As long as there is no coupling to an ice sheet model, it is standard for an AOGCM to operate with a fixed surface topography over land. As far as I can see it, this has nothing to do with technical requirements of the snow pack model as described here. It would be interesting to describe instead if and how the snow-pack model differs from other GCMs and from the MAR model, which I suspect has a full physical solution to the problems you are describing. *The snow model used in MAR is much more sophisticated than ours, but MAR has the same issue due to the lack of ice-sheet dynamics. We don't provide a compared description of both snowpack models since this would not help interpreting our results. In the ablation zone of the GrIS (where the annual SMB is negative), all the snow that falls during the cold season melts during summer, and some of the underlying ice also melts, which is compensated by approximately the same amount of advected ice (if the ice sheet is in quasi-equilibrium). To represent the negative SMB in MAR, a reservoir of ice (20 m thick) has been introduced in the model in the ablation zone (Lefebre et al, 2004), and this ice partly melts during the simulation. In CNRM-CM5, we use a similar approach, except that our reservoir consists of snow rather than ice (same latent heat of fusion as snow per kg, and only the changes in the mass of snow matter, not the snow depth). Since we run very long simulations with CNRM-CM5, our snow reservoir needs to be "huge", to make sure that is not entirely depleted even after 1000 years of simulation. Our method also ensures that the amount of water (liquid + frozen) in our climate model system is conserved.

L81-83: Since there is no ice-dynamical process in this model at all, it seems strange to evoke the idea of a calving flux. *It's actually a pseudo-calving flux, it corresponds to the calving flux from the GrIS that would be simulated if the dynamics of the ice-sheet were represented.

I think it would be far simpler to say that all precipitation over ice-covered land is equally distributed over the ocean north of 60N, while the snow pack evolution is calculated diagnostically, without contribution to the mass budget. It should be clarified that the instantaneous relocation of this mass (freshwater?) as an additional forcing does not have any influence on the ocean response. *The correction is applied on the snow reservoir, not precipitation, and the description of that process has to be consistent with what is really done in the paper. Hence the statement in the paper is maintained "To avoid unrealistic snow accumulation on the GrIS and an associated decrease in the modelled sea level, a pseudo-calving flux is computed at every time step from the spatially integrated snow reservoir excess over the GrIS and is distributed over the ocean north of 60°N."

L86: A resolution of 40-50 km is still relatively low compared to the resolution of state-of-the-art regional climate models (MAR at 15km, RACMO at 11km). This should mentioned here. *We choose not be mention this here, but later in the paper (lines 186-188), where we suggest that the still relatively coarse resolution of the model may hamper the simulation of the spatial variability of surface melting, which meets your point.

L120: If model bias and climate change signal are combined, how do we tell them apart? Is there maybe another experiment that could separate these two factors? *In the case of global forced atmospheric simulations (with SST), it is, to some extent, possible to disentangle model biases from climate change signals. However, even in this idealized context, the phases of the large variability structures (NAO, PDO, etc.) can differ from observations (this would be the case even if the global model was 'perfect'). This aspect impacts estimates of climate change signals. In the case of

a free global coupled ocean-atmosphere model like CNRM-CM5-2, it is even more difficult to separate model biases from climate change signals, since the ocean can produce very low frequency variability (centennial), of the order of the climate change signal itself. In observations, it is however possible, especially in high-variability areas to disentangle long term climate change from e.g. multi-decadal variability (detection).

L124: Is this discussion really important for the GrIS? Consider discussing the biases for Greenland in more detail instead. *Figure 1 has been redone with a focus on the Arctic. The analysis of biases in our global simulation has thus been removed.

L169: Is it elevation or surface slope that has an important impact on precipitation amounts? Clarify. *Our statement oversimplifies the underlying processes of accumulation, which depends on elevation, slope and the characteristics of the atmospheric flow. More over it could not be supported by the figures, since we do not provide elevation. Hence we just mentioned that the simulated patterns of accumulation are similar in MAR and NPS-0k.

L178: Clarify if this masking includes ice caps and glaciers in the periphery of the Greenland ice sheet. *The high spatial resolution Greenland mask from GADM does not include ice caps and glaciers.

L185: You attribute most of the underestimation of melt to the albedo limit. Why is that limit in place? *We use a global model, and some tuning parameters reflect a compromise to globally limit model biases in its representation of the snowpack (especially seasonal).

Are there other shortcomings of the snowpack model worth mentioning? How does the snowpack model compare in complexity and included processes to the one in MAR? *In the §2.1 we indicated that the snowpack is represented by the one-layer snow scheme of Douville et al. (1995). This model has been much used at Météo-France in climate modelling and numerical weather prediction until a recent transition (for CMIP6) to the new ISBA-ES model (no reference available yet). We added more information about

this snow scheme and that of MAR (SISVAT) in the paper.

If resolution is an important limitation, how does the model compare to low resolution versions of MAR (Franco et al., 2012). *Even if our model resolution on Greenland is close to that of Franco et al (2012), another big difference between our global model and their model is the lateral constraint !

L194: Could add a few references after "Greenland" as a reminder. *We actually removed this sentence, which is quite a general statement out of place in this part of the paper.

L191: I strongly disagree with this statement. The model is clearly not reproducing the melting well and therefore shows considerable shortcomings to represent the SMB. The statement is in clear contradiction to the description L184 and L312. *Indeed we underlined the underestimation of the melting, however note that the simulated equilibrium line, which separates the accumulation zone from the ablation zone, is rather realistic compared to MAR (Fig. 4k-l) and the correlations between the NAO index and the GrIS-averaged melting as SMB are consistent with MAR (Table hereafter).

L195: Could you please clarify if the NAO index is here calculated based on the normalised PC as described at line L156? In other words, is the NAO index definition the same for the ERA-based correlation with MAR SMB as the CNRM-CM5.2 correlation with CNRM-CM5.2 SMB? *The NAO index was calculated in the same way for NPS and ERA-Interim, namely from the normalized first PC of the detrended sea level pressure.

L215: Are "changes in precession" meant compared to pre-industrial or to other times during the Eemian? *Theses changes are meant wrt preindustrial, as now stated in the paper.

L246: Could you find a better word instead of "node"? This is the first time this term is used. Maybe 'region'? *This is standard in the community to refer to both centers of

action of NAO (Islandic Low and Azores High).

L310: Again, I think this statement may be true for accumulation, but clearly not for melting. *Already answered previously.

L317: There is "another hand" missing in this sentence or somewhere in the following. *Done. This sentence has been moved to §3.3.

L331: Not sure what "nibbled" means, please revise. Interesting to speculate on the impact of the Greenland ice sheet during the Eemian, extend if possible. *There is little literature about this. Ideally, an ice sheet model should be used to investigate this aspect more in depth.

L344: This final statement may raise the suspicion that the findings in this paper are not yet established to be robust and may be subject to change. Maybe just a question of formulation. Revise. *Thanks for this suggestion. Done.

Figure 2 Precipitation is defined positive. Maybe adjust the colour scale accordingly? *We adjusted the colour scale to follow this recommendation.

Figure 4 This figure clearly shows that ablation and SMB are very poorly represented in NPS-0k. Can you show the sublimation E subtracted from P to get accumulation C in the top panel (maybe as a supplement)? It seems to have a large impact on the resulting C. It also seems to have large spatial variability. Is that expected? *According to your suggestion we have evoluted the paper taking account SMB components rather the SMB itself. Now Figure 4, shows seasonal (DJF and JJA) means of accumulation, melting and SMB over Greenland from NPS-0k and MAR. Inside Greenland, the SMB of NPS-0k are, like for annual averages, a little noisy. The figure hereafter show for NPS-0k and MAR, the direct sublimation that are removed from precipitation. Inside Greenland, the accumulation of NPS-0k is underestimated (Table 2) due to strong direct sublimation however this does not vary much in space.

— References — Franco, B., Fettweis, X., Lang, C., & Erpicum, M. (2012). Impact

of spatial resolution on the modelling of the Greenland ice sheet surface mass balance between 1990–2010, using the regional climate model MAR. The Cryosphere, 6(3), 695–711. http://doi.org/10.5194/tc-6-695-2012

Hanna, E., Cropper, T. E., Hall, R. J., & Cappelen, J. Greenland Blocking Index 1851–2015: a regional climate change signal. International Journal of Climatology, 36(15), 4847-4861, 2016. Hanna, E., Jones, J. M., Cappelen, J., Mernild, S. H., Wood, L., Steffen, K., & Huybrechts, P.: (2013). The influence of North Atlantic atmospheric and oceanic forcing effects on 1900–2010 Greenland summer climate and ice melt/runoff. International Journal of Climatology, 33(4), 862-880, 2013 Lefebre, F., Fettweis, X., Gallée, H., Van Ypersele, J. P., Marbaix, P., Greuell, W., & Calanca, P.: Evaluation of a high-resolution regional climate simulation over Greenland. Climate dynamics, 25(1), 99-116, 2005. Mysak, L.A. : Glacial inceptions: Past and future, Atmosphere-Ocean, 46:3, 317-341, DOI: 10.3137/ao.460303, 2008. Roche, Didier & Dumas, Christophe & Ritz, Catherine & Goosse, Hugues. Transient simulation of the last inception (128 to 115ka BP) with a coupled climate - cryosphere model. 12. 12632, 2010.

[Figure]

**Fig. 1.**

[Figure]

**NPS-0k  JJA (2035-2315)**

[Figure]

**NPS-0k  JJA (2035-2315)**

[Figure]

[Figure]

**MAR JJA (1980-1999)**

[Figure]

**MAR JJA (1980-1999)**

[Figure]

**Fig. 2.**

[Figure]

|  | Accumulation | | Melting | | SMB | |
|---|---|---|---|---|---|---|
|  | DJF | JJA | DJF | JJA | DJF | JJA |
| **MAR** | **-0.21** | **0.54** | **-** | **0.37** | **-0.21** | **0.40** |
| NPS-0k | -0.22 | 0.48 | - | 0.51 | -0.22 | 0.62 |
| NPS-115k | -0.11 | 0.43 | - | 0.56 | -0.11 | 0.62 |
| NPS-130k | -0.04 | 0.48 | - | 0.43 | -0.04 | 0.56 |

Seasonal (DJF and JJA) correlations between accumulation, melting and SMB averaged on GrIS and the NAO index.

**Fig. 3.**

**BBfC4  DJF (2035-2315)**

DJF: Grid averaged snow and ice sublimation flux                    m/yr

80N

70N

60N

60W        30W

-1.  -.7  -.5  -.3  -.1   0.   .1   .3   .5   .7   1.

Min=  -0.01                    Max=   0.10

**MAR DJF  (1980-1999)**

Sublimation/evaporation (sub-pixel 1)                    mWE/yr

80N

70N

60N

60W        30W

-1.  -.7  -.5  -.3  -.1   0.   .1   .3   .5   .7   1.

Min=  -0.13                    Max=   0.26

**BBfC4  JJA (2035-2315)**

JJA: Grid averaged snow and ice sublimation flux                    m/yr

80N

70N

60N

60W        30W

-1.  -.7  -.5  -.3  -.1   0.   .1   .3   .5   .7   1.

Min=  -0.02                    Max=   0.08

**MAR JJA  (1980-1999)**

Sublimation/evaporation (sub-pixel 1)                    mWE/yr

80N

70N

60N

60W        30W

-1.  -.7  -.5  -.3  -.1   0.   .1   .3   .5   .7   1.

Min=  -0.79                    Max=   0.36

**Fig. 4.**